# Nutrition in Ultra-Endurance: State of the Art

**DOI:** 10.3390/nu10121995

**Published:** 2018-12-16

**Authors:** Pantelis T. Nikolaidis, Eleftherios Veniamakis, Thomas Rosemann, Beat Knechtle

**Affiliations:** 1Laboratory of Exercise Testing, Hellenic Air Force Academy, 13671 Dekelia, Greece; pademil@hotmail.com; 2Exercise Physiology Laboratory, 18450 Nikaia, Greece; 3Department of Nutrition and Dietetics, Technological Educational Institute, 72300 Sitia, Greece; veniamakise@googlemail.com; 4Institute of General Practice and for Health Services Research, University of Zurich, 8091 Zurich, Switzerland; thomas.rosemann@usz.ch; 5Medbase St. Gallen Am Vadianplatz, 9001 St. Gallen, Switzerland

**Keywords:** cycling, fluid overload, exercise-associated hyponatremia, limb swelling, swimming, ultra-marathon

## Abstract

Athletes competing in ultra-endurance sports should manage nutritional issues, especially with regards to energy and fluid balance. An ultra-endurance race, considered a duration of at least 6 h, might induce the energy balance (i.e., energy deficit) in levels that could reach up to ~7000 kcal per day. Such a negative energy balance is a major health and performance concern as it leads to a decrease of both fat and skeletal muscle mass in events such as 24-h swimming, 6-day cycling or 17-day running. Sport anemia caused by heavy exercise and gastrointestinal discomfort, under hot or cold environmental conditions also needs to be considered as a major factor for health and performance in ultra-endurance sports. In addition, fluid losses from sweat can reach up to 2 L/h due to increased metabolic work during prolonged exercise and exercise under hot environments that might result in hypohydration. Athletes are at an increased risk for exercise-associated hyponatremia (EAH) and limb swelling when intake of fluids is greater than the volume lost. Optimal pre-race nutritional strategies should aim to increase fat utilization during exercise, and the consumption of fat-rich foods may be considered during the race, as well as carbohydrates, electrolytes, and fluid. Moreover, to reduce the risk of EAH, fluid intake should include sodium in the amounts of 10–25 mmol to reduce the risk of EAH and should be limited to 300–600 mL per hour of the race.

## 1. Introduction

An ultra-endurance race may refer to running, cycling, swimming, cross-country skiing or a multi-sport event such as triathlon. The criterion to define these modes of exercise as ultra-endurance is to last at least six hours [1]. On the other side of the spectrum of duration, an ultra-endurance race may last even days or weeks. In these races, athletes usually compete for time, i.e., they attempt covering a given distance in the least time; however, other formats of races exist too, e.g., they might compete for distance where they aim to cover a maximum distance for a given time (e.g., 12 h run). Considering the long duration of these races, an ultra-endurance athlete faces sport-specific nutritional issues.

A major nutritional concern is the high energetic demands accompanying the physiological stress of exercise. Ultra-endurance exercise induces an energy deficit (negative energy balance, i.e., energy expenditure outscores energy intake), which in turn may lead to an energy depletion [2]. The increased exercise metabolism induces heat production triggering sweating rates at a risk of hypohydration (2%–5% of body water loss). In this case, hydration should be optimized in order to avoid medical issues, such as hyponatremia and heat illness [3]. For instance, an increased fluid intake compared to fluid loss might lead to exercise-induced hyponatremia—which is associated with hyponatremic encephalopathy, altered mental status, collapse, seizure, coma, and death—during prolonged races [3].

In general, ultra-endurance athletes adopt nutritional strategies similar to endurance athletes; nevertheless, they might present unique nutrition-related characteristics. For instance, a higher prevalence of vegan/vegetarian diets have been observed in ultra-marathon than in marathon runners [4]. Furthermore, ultra-endurance and endurance athletes differ in age with the former being older than the later, as well as in anthropometric and training characteristics.

Despite the increasing participation in ultra-endurance sports during the last years, the scientific interest in these sports remains relatively small with regards to other sports [5,6]. Accordingly, only a few reviews have been ever conducted on nutrition and ultra-endurance [2,7,8], and these focused mostly on walking and running events without covering the whole spectrum of ultra-endurance sports including other sports disciplines such as swimming and cycling. Considering the large number of athletes that increasingly participates each year in ultra-endurance sports, a comprehensive review of nutritional aspects concerning these athletes would be of great practical importance for a wide audience. Therefore, the aim of the present study was to review all published original articles in this topic focusing on energy deficit and dysregulation of fluid metabolism (Figure 1).

## 2. Energy Deficit

The energy balance refers to the difference between energy expenditure and energy intake [9,10,11]. In turn, the energy expenditure is assessed using triaxial accelerometry [10], indirect calorimetry [12] or the energy equivalent of oxygen that can be calculated from heart rate recordings during the race [9,11]. Energy intake is estimated by nutritional analysis of all drinks and foods contents consumed during a race [9,10,12].

### 2.1. Energy Balance in Race

The long duration of an ultra-endurance race implies increased exercise-induced energy expenditure [13]. To balance the increased energy expenditure, an optimal nutrition should ensure adequate energy intake [14]. So far, research has concluded that endurance and ultra-endurance athletes do not consume sufficient food and drinks resulting in a negative energy and fluid balance during the race, as shown by their reduced post-race body mass and body fat percentage [15,16]. For example, a 54-km mountain ultra-marathon induced a negative energy balance of ~3700 kcal [17]. In addition, a negative energy balance might be related to other undesired nutritional practice and physiological responses, such as a low consumption of antioxidant vitamins [18] and a pronounced drop in insulin-like growth factor 1 [19].

The performance in an ultra-endurance sport depends on an optimal nutrition during the race [20]. An inadequate nutrition results in a negative energy balance which in turn might impair performance. For instance, a negative relationship between caloric intake and finish time (i.e., the more the intake of calories from nutrition, the faster the race time) has been shown in a 384-km cycling race [21]. An ultra-endurance race induces an energy deficit [22]. Figure 2 presents examples of energy balance in swimming, cycling, running, and triathlon. Considering these data, it might be supported that swimming induces a larger energy deficit—likely due to the limited access to food and drink during a race—than cycling and running. In races with duration of more than 24 h, multi-sports (e.g., triathlon) and cycling induce the larger energy deficit, 3× more than the corresponding value in running.

### 2.2. Change of Body Mass during Race

Performing an ultra-endurance race can decrease body mass in levels that may reach to losses greater than 5% of their starting body weight [42] (Figure 3). For example, a 24-h running ultra-marathon, where the covered distance ranged from 122 to 208 km, induced a decrease of body mass by 1.7% [43], whereas a ~230-km ultra-marathon resulted in a decrease of 1.0%–2.5% [44]. In addition, an 800-km Antarctic race (14–28 days) resulted in a reduction of body mass and lean mass by 8.3 kg and 2.0 kg, respectively [45]. It has been shown that the decrease of body mass is located mostly in the lower trunk [36]. Whether the loss of body mass reflects to a fat mass, skeletal muscle mass or a combination of them as well as fluid loss depends on the duration of race [46]. For instance, the largest decrease of skeletal muscle mass was observed in events such as 24-h swimming, 6-day cycling or 17-day running (Figure 3). It might be assumed that a concentric exercise (cycling) rather induces a loss of fat mass [47], whereas an eccentric exercise (running) rather causes a loss of skeletal muscle mass [48]. Furthermore, a reduction in both fat mass and skeletal muscle mass has been observed in runners [48]. For swimmers participating in a 12-h indoor pool, no change in body mass, fat mass or skeletal muscle mass was shown [49], whereas for male open-water ultra-swimmers, a loss in skeletal muscle mass was reported [50]. Since swimming is primarily a concentric exercise, the loss in skeletal muscle mass might be dependent on the duration of the race, too. Although most of the research conducted in ultra-endurance races reported a decrease of body mass across the races, inconsistencies in the findings regarding changes in fat and skeletal muscle mass among studies [46,47,49] should be attributed to race specific characteristics (e.g., duration and mode of exercise) and inter-individual variability (i.e., responders and non-responders) [15].

In contrast to the abovementioned studies that observed decreases in body mass, fat mass, and skeletal muscle mass, other research reported an increase in body mass and skeletal muscle mass during ultra-endurance races [39]. An explanation of the increase in body mass might be the occurrence of fluid overload (see next section). Moreover, an increase in skeletal muscle mass might be shown when anthropometric methods (skin-fold thicknesses and limb circumferences) were used to quantify body composition. However, total body water seems to increase by ~1.5 L [13]. In summary, ultra-endurance athletes lose ~0.5 kg in body mass and ~1.4 kg in fat mass.

### 2.3. Energy Balance in Training

Considering the duration of the training units, the training of ultra-endurance athletes included “endurance” rather than “ultra-endurance” exercise. For instance, a master swimmer trained 3 h per day (15–70 km weekly) prior to a 78-km open water swim [57]. Also, 100-km ultra-marathon runners reported ~8 h running weekly [58]. So far, in contrast to the abovementioned observations of larger energy expenditure than intake during ultra-endurance races [42,43,44,45], a negative energy balance has not been reported during training of ultra-endurance athletes, and thus, it might be assumed that changes of body mass, fat, and skeletal muscle mass during training might be trivial.

## 3. Hypohydration, Fluid Intake, and Fluid Overload

A main concern of ultra-endurance athletes is hypohydration, which can either be induced prior to exercise or develop during exercise (exercise-induced hypohydration). The exercise-induced hypohydration decreases aerobic performance and increases body temperature, heart rate, perceived exertion, and reliance on carbohydrate (CHO) as a main fuel source [59]. Excessive levels of hypohydration can be prevented by fluid replacement during exercise, and adding Na^+^ in sport drinks is expected to increase a sense of thirst and reduce fluid losses, as well as prevent symptoms of hyponatraemia [60,61,62]. Moreover, when fluid consumption is adequate, the body mass can be unchanged [63]. On the other hand, when the fluid consumption is in excess (overload) body mass increases and plasma sodium concentrations decrease by increasing the risk of exercise-associated hyponatremia (EAH) [64]. For instance, an increase in body mass of 8 kg within the first three days of a Deca Iron ultra-triathlon covering 38 km of swimming, 1800 km of cycling, and 422 km of running in ~13 days has been reported [53]. A post-race increase in body mass usually is accompanied by increases in anthropometric characteristics, such as skin-fold thicknesses and limb circumferences [53]. A common trait of these studies was that they examined races conducted in hot temperature where most probably fluid consumption was rather augmented. Nonetheless, an increase in skinfolds’ thickness has been observed even in cases of race-induced body mass loss [42]. An athlete who presented a loss in body mass after a Triple Iron ultra-triathlon had considerable swelling of the feet [39].

Likely, the increase in body mass, skin-fold thicknesses, and limb circumferences might be due to an increase in body water [65]. An increase in total body water in ultra-endurance athletes has been shown in several studies [42,66]. An explanation of the increase in total body water might be an increase in plasma volume, which might be due to sodium retention due to an increased activity of aldosterone [67]. An increase in plasma volume is related to an increase in the potassium-to-sodium ratio in urine that suggests that an augmented activity of aldosterone may induce retention in both sodium and fluid during an ultra-endurance race [68]. In a multi-stage race over seven days, total mean plasma sodium content augmented and was the main cause of the increase in plasma volume [69].

In addition to the abovementioned acute physiological responses to ultra-endurance exercise, fluid overload can also induce an increase in limb volume. A relationship between changes in limb volumes and fluid intake has been documented recently [70]. Since neither renal function nor fluid regulating hormones were related with the changes in limb volumes, fluid overload might be considered as the most possible cause for increase in both body mass and limb volumes. An association between an increased fluid intake and swelling of the feet in ultra-marathoners has been observed, too [71].

Fluid overload can cause EAH, which is defined as a serum sodium concentration ([Na^+^]) <135 mmol.L^−1^ during or within 24 h of exercise [72]. The first time EAH was described in the scientific literature was in 1985 in ultra-marathoners in South Africa as being due to “water intoxication” [73]. The causes for EAH in endurance athletes are: (i) overdrinking due to biological or psychological factors; (ii) inappropriate secretion of the antidiuretic hormone (ADH), in particular, the failure to suppress ADH-secretion in the face of an increase in total body water; and (iii) a failure to mobilize Na^+^ from the osmotically inactive sodium stores or alternatively inappropriate osmotic inactivation of circulating Na^+^ [72]. Since we have a poor understanding of the mechanisms of the factors (i) and (iii), it is subsequent that to prevent EAH requires that athletes be encouraged to avoid overdrinking during exercise. According to a study by Barr and Costill [74], the amounts of fluid lost can vary from 600 to 2000 mL·h^−1^. Such amounts of intake are unrealistic to be consumed by an individual as the maximum fluid intake can be limited to 300–600 mL·h^−1^ that in some situations can only cover 50% of the body weight lost [75]. Greater intakes than 300–600 mL·h^−1^ may lead to gastrointestinal discomfort, and moreover, when the exercise intensity and the sweat rate decreases may overhydrate the athlete and lead to hyponatremia situation that more usually occur in female athletes [72,76,77].

Exercise-associated hyponatremia is the most commonly occurring health condition of ultra-distance exercise and is typically from excessive intake of hypotonic fluids [78]. The main cause of EAH is the habit of over-drinking during a race occasionally combined with inadequate sodium intake [79]. Athletes suffering by EAH consume the double quantity of fluids during a race compared to athletes without EAH [80]. Another aspect of EAH is its relationship with performance level, as it has been shown a higher fluid overload for slower athletes [81]. On the contrary, faster athletes do not develop EAH even if they drink more than their slower counterparts [82].

It has been observed that the environmental conditions influence the occurrence of EAH. Frequently, EAH can be reported in ultra-endurance races held in extreme cold or extreme heat [64]. On the contrary, EAH is relatively uncommon in temperate climates [83]. Nonetheless, the occurrence of EAH in ultra-marathoners [84] is similar as in marathoners [85].

The occurrence of EAH might also be varying by discipline (Figure 4). Although EAH was largely frequent in ultra-swimming [77] and ultra-running [86], its prevalence was low or even absent [87] in ultra-cycling. An interpretation of this finding might be that cyclists have the opportunity to individually drink by using their own drink bottles on the bicycle. Moreover, the duration of an ultra-endurance race appears to augment the risk for EAH. The highest prevalence of EAH has been observed in Ironman triathlons [88], Triple Iron ultra-triathlons, and ultra-marathons covering 161 km [86].

With regards to fluid intake during an ultra-endurance race, an ad libitum strategy—close to the range 300–600 mL·h^−1^—has been recommended to decrease the risk of EAH and optimize plasma sodium concentration [99]. An intake of 300–400 mL·h^−1^ has been suggested to decrease the risk for EAH, whereas ~400 mL·h^−1^ would optimize serum sodium concentration in a 4 h walk [99]. Furthermore, a fluid intake of ~400 mL·h^−1^ might decrease the risk of EAH in a 161-km race in the cold [79]. In all situations, the volume of fluid consumption depends from the individual’s level of thirst, the gastrointestinal comfort, and the source and the type of fluid and the temperature of the drink [100,101]. Although it would be reasonable to assume that the consumption of sodium supplementation would help preventing EAH, recent research showed that this supplementation might not be necessary. For instance, research on Ironman triathlon indicated that serum sodium concentration could be preserved even without supplementation [102].

## 4. Nutritional Aspects in Ultra-Endurance Athletes

To finish an ultra-endurance race, optimal energy and fluid intake is necessary [103]. The relevant contribution of CHO, fat, and protein to the total energy intake can be seen in Figure 5, according to which the energy intake relies mostly on CHO following by lipids and protein. Nevertheless, under particular environmental conditions such as cold temperature, a relatively low CHO intake might be practiced. For instance, the caloric intake was 23.7% from CHO, 60.6% fat and 15.7% protein in an 800-km ultra-endurance Antarctic race [45].

### 4.1. Intake of Carbohydrates

Carbohydrates are the main energy source in ultra-endurance athletes like in the general population [14]. When the nutritional intake was analyzed in ultra-endurance athletes, the largest percentage was observed for CHO. Ultra-endurance athletes consume ~68% of energy intake as CHO. Ultra-endurance athletes consume more CHO (90 g·h^−1^) than endurance athletes (60 g·h^−1^) and this difference should consider body mass and training status [107]. The CHO intake has been reported 31 g·h^−1^ in an ultra-marathon race [108], 42 g·h^−1^ in the Marathon des Sables [109], 56 g·h^−1^ in 4254 km ultra-marathon over 78 days [110], and 83 g·h^−1^ participating in seven open water swimming races of 15–88 km distance and 3–12 h duration [111]. In addition to the quantity, the quality of CHO is related to race time, e.g., an optimal glucose-to-fructose ratio during exercise might increase the exogenous CHO-oxidation and prevent gastrointestinal distress [112]. Strategies such as CHO-loading can be applied to maximize the storage of muscle and liver glucose. It is well documented that such strategies can improve the duration and performance in ultra-endurance sports.

Three well known CHO-loading protocols are able to super-compensate the glycogen stores. The first protocol, long taper or classic protocol was originally developed in the late 1960s by Hultman and colleagues [113] when they started examining fuel utilization and enzyme activities in the skeletal muscle in response to exercise. The evidence of the studies has shown that the capacity for a prolonged exercise in a moderate intensity was associated and determined by the pre-exercise muscle glycogen stores. Based on these findings, the classic protocol involved a 3–4 days glycogen storage depletion (“depletion phase”), undertaking a high-intensity exercise and low CHO intake (10%) followed by a 3–4 days high CHO intake “loading phase” and low-intensity exercise (exercise taper) overloading body glycogen storage. The second protocol, which was a modified six-day-long protocol, aimed to deplete the glycogen storage and steadily increase the CHO intake and decrease the physical activity without the severe and stressing for the body glycogen depletion phase of the classic protocol [114]. Sherman and colleagues’ protocol [114] was more practical for the athletes, more suitable before competition, needed less time to accomplish, avoided the fatigue and the extreme glycogen depletion, and had slightly less but still great glycogen overload as a result with the classic protocol. The third and most recent one had rapid CHO loading [115]. A modified protocol that lasts in a range of 36 to 48 h prior to the event and could highly increase the muscle and liver glucose storage and achieve similar result as the previous protocols in less time and less stress for the athlete. In this study [113], glycogen storage maximized after one and three days of resting and a high CHO diet (10 g·kg^−1^ of body weight). The abovementioned study concluded that optimal refueling can be achieved within 36 to 48 h of the last exercise bout, as long as the athlete consumed high CHO (70% of energy) and rest during these days. Carbohydrates loading is well known to enhance exercise performance not by increasing the overall speed of running but by maintaining the race pace during the last period of the run in exercise lasting more than 90 min by about 20%. On the other hand, high CHO intake and rest had, as a result, an increase in body weight by up to 2 kg [116].

It should be highlighted that most of the body of the literature concerning CHO intake strategies referred to endurance rather than to ultra-endurance races [117]. For instance, another strategy was the so-called “train low run high”; where the runner consumed low CHO during training and high CHO in race [117]. Nevertheless, the efficiency of low CHO intake has been debated, since the periodic consumption of low CHO did not reveal superior effects on performance of endurance athletes compared to high CHO intake [40]. The recently developed concept of periodized nutrition has been defined as “planned, purposeful, and strategic use of specific nutritional interventions to enhance the adaptations targeted by individual exercise sessions or periodic training plans, or to obtain other effects that will enhance performance longer term” [117]. Unfortunately, the CHO intake in ultra-endurance athletes has been studied mostly during races, whereas scarce information exists with regards to nutritional behavior during training.

### 4.2. Intake of Fat

Compared with endurance sports that rely mostly on the metabolism of CHO, ultra-endurance sports present increased demands in fat intake considering the length of the races and larger amounts of fat than CHO stored in the human body [118]. “Fat-adapt” has been developed as a pre-race nutritional strategy to increase fatty acid oxidation, attenuate the depletion of glycogen, and ameliorate performance [118,119]. This strategy included high fat (60%–70%) and low CHO (15%–20%) intake for 5–10 days prior to the race to enhance fat utilization [118]. Considering that an ultra-endurance race was performed at a sub-maximal intensity, an increased intake of fat would not be detrimental for performance [119]. For example, a 6-day fat-adapt diet consists of 2.6 g·kg^−1^·day^−1^ CHO and 4.6 g·kg^−1^·day^−1^ fat [120] or 70% fat [121], whereas a 5-day fat-adapt diet includs 2.5 g·kg^−1^·day^−1^ CHO and 4.3 g·kg^−1^·day^−1^ fat [122].

The role of fat consumption has been studied during training [115,123] and race [110,124], where a nutritional strategy in its intake was to increase fatty acid oxidation for a given exercise intensity and meet the increased caloric demands, respectively. It has been shown that high pre-race fat consumption induces an augmentation of intramyocellular lipids in ultra-endurance athletes, an adaptation that might enhance performance [115]. Also, a case study of a rower who consumed a diet with high fat for a fortnight showed an increase in ultra-endurance performance [123]. Considering fat as an important fuel for human body, it would be assumed that an increased fat intake during an ultra-endurance race might improve race time. However, field-controlled studies have been conducted. In an ultra-marathoner (6-day ultra-marathon) who consumed 34.6% of fat in his daily food intake [124], body fat was lowered in the first two days of the race, and thereafter, remained unchanged, and performance slowed down after the first two days. Ultra-endurance athletes consume ~19% of ingested energy from fat, which is more than energy intake from protein. Furthermore, it has been shown that the consumption of fat might vary by race duration with athletes participating in longer races showing a higher intake of fat [110].

### 4.3. Intake of Protein

Considering the consumption of protein, it has been shown that this macronutrient consists ~19% of energy intake during racing in ultra-endurance cyclists during the “Race across America” [35]. An explanation of this elevated percentage might be that athletes increase the relative consumption of amino acids in order to prevent the loss in skeletal muscle mass. However, the effectiveness of amino acids has not been documented yet; for instance, the consumption of amino acids did not influence skeletal muscle damage [125]. In addition, studies have shown that protein intake more than 2 g·body weight·day does not increase athletic performance.

### 4.4. Intake of Ergogenic Supplements, Vitamins, and Minerals

The consumption of vitamin and minerals during training and competition is a common nutritional habit in ultra-endurance [126]. Thus, it is no surprise that the effect of vitamins, minerals, and other ergogenic supplements has been studied in ultra-endurance athletes [127]. It should be highlighted that intake of vitamin C and E could even blunt training adaptations [128]. A survey in long-distance triathletes reported that the majority used vitamin supplements during training with vitamin C being the most popular, followed by vitamin E and multivitamins [127]. In addition, this survey showed that the primary cause for the use of these supplements was to decrease the risk of getting cold. Although the beneficial role of vitamins and minerals for health is well documented, they do not improve performance [129]. A comparison between athletes that were administered vitamins and minerals for a four-week pre-race period and those that were not provided with supplements did not find any difference in race time in a 200-km multi-stage ultra-marathon (“Deutschlanlauf 2006”) [12]. In another example (Triple Iron ultra-triathlon), a comparison between athletes with and without regular pre-race consumption of vitamins and minerals found no difference in race time, too [130]. The consumption of supplements did not affect physiological aspects related to performance [131]. For instance, the consumption of chronic probiotics in the Marathon des Sables did not alter extracellular Hsp72 [129]. Furthermore, the intake of antioxidant vitamins by ultra-endurance cyclists did not induce alterations in their endogenous antioxidant defending exercise-induced reactive oxidative stress [131].

An iron status deficiency is most likely to occur among the micronutrients as inadequate iron status can impair exercise performance through suboptimal levels of hemoglobin and also through changes in the muscle including reduced myoglobin and iron-related enzymes [132]. Exercise has a greater influence as it alters the plasma volume and the functions response to exercise stress and fatigue, causing reduction of blood hemoglobin concentrations that result from the expansion of plasma volume in response to endurance and ultra-endurance exercise. This deficiency known as sports anemia does not impair performance [133]. It is hard to conclude if this deficit is a true deficit in iron status that impairs health and performance or sports anemia. Monitoring the athlete’s nutrition habits and providing with adequate doses of bioavailable iron in foods would help prevent the true deficit. Ferritin levels <20 ng·mL^−1^ are a reliable indicator to diagnose true anemia. In cases where iron status is at a low level, supplements may need to consumed by the athlete with the assistance of a doctor to refill the lost amount, and avoid excessive amounts of iron intake as it may lead to hemochromatosis.

## 5. Conclusions

In summary, ultra-endurance athletes do not consume enough energy to counterbalance the race-induced energy deficit, which results in decrease of body mass. Optimal pre-race nutritional strategies should aim to increase fat utilization during exercise, and the consumption of fat-rich foods may be considered during the race as well as the CHO, electrolytes, and fluid. As the length of ultra-endurance races increase, athletes increase fluid intake which increases the risk for exercise-associated hyponatremia and limb swelling. To reduce this risk, fluid intake should be limited to 300–600 mL per hour.

## Figures and Tables

**Figure 1 nutrients-10-01995-f001:**
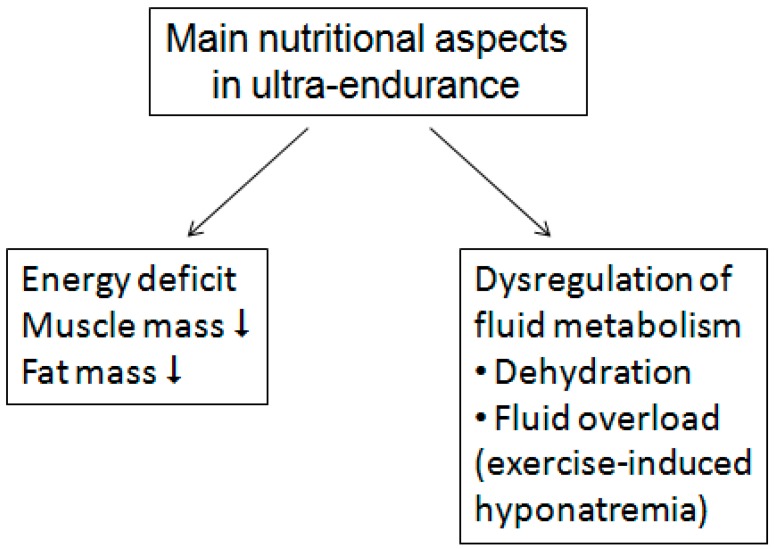
Main nutritional aspects in ultra-endurance. ↓ = decrease.

**Figure 2 nutrients-10-01995-f002:**
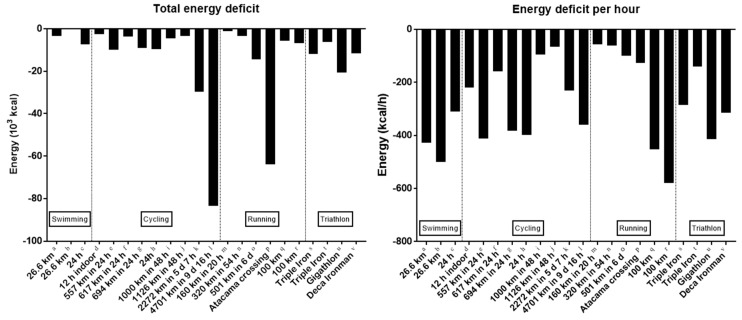
Energy balance in ultra-endurance athletes in swimming, cycling, running, and triathlon. a = [23], b = [24], c = [25], d = [26], e = [27], f = [28], g = [29], h = [30], i = [31], j = [24], k = [32], l = [11], m = [33], n = [34], o = [35], p = [36], q = [37], r = [38], s = [39], t = [40], u = [41], v = [42].

**Figure 3 nutrients-10-01995-f003:**
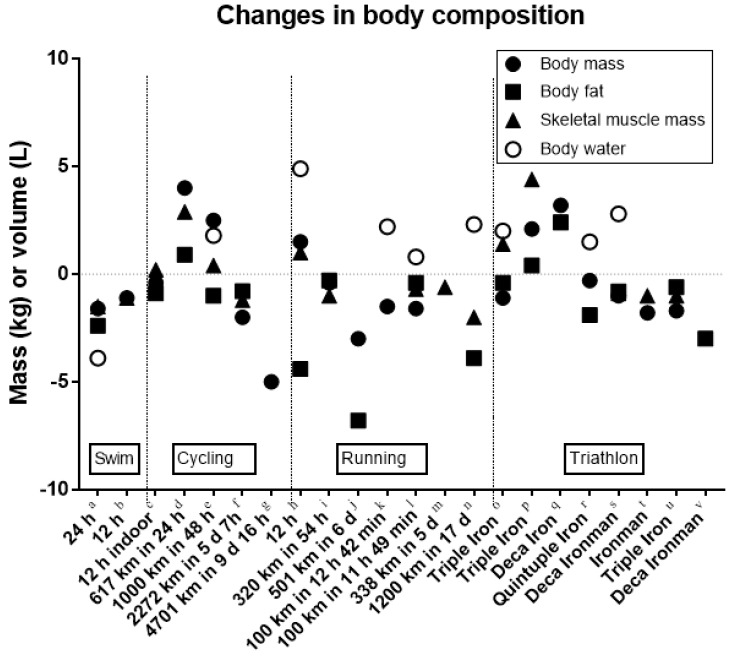
Change in body composition in ultra-endurance athletes competing in swimming, cycling, running, and triathlon. a = [25], b = [24], c = [26], d = [28], e = [32], f = [33], g = [11], h = [51], i = [34], j = [35], k = [37], l = [46], m = [48], n = [52], o = [39], p = [40], q = [53], r = [13], s = [42], t = [54], u = [55], v = [56].

**Figure 4 nutrients-10-01995-f004:**
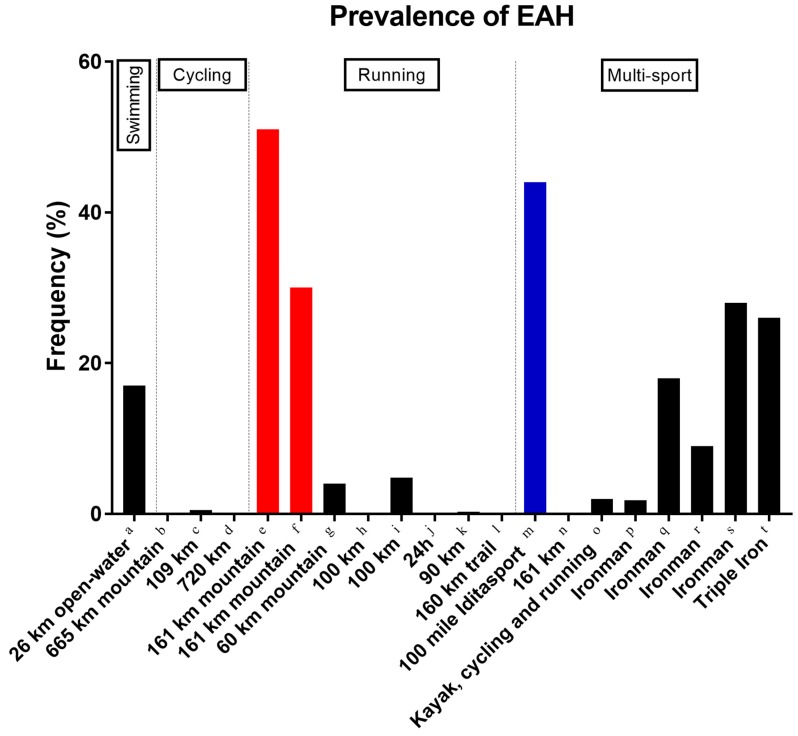
Prevalence of exercise-associated hyponatremia (EAH) in ultra-endurance athletes competing in swimming, cycling, running, and multi-sports disciplines. Red and blue highlight hot and cold environment. a = [77], b = [89], c = [90], d = [87], e = [64], f = [86], g = [83], h = [91], i = [82], j = [92], k = [73], l = [63], m = [79], n = [93], o = [94], p = [95], q = [88], r = [96], s = [97], t = [98].

**Figure 5 nutrients-10-01995-f005:**
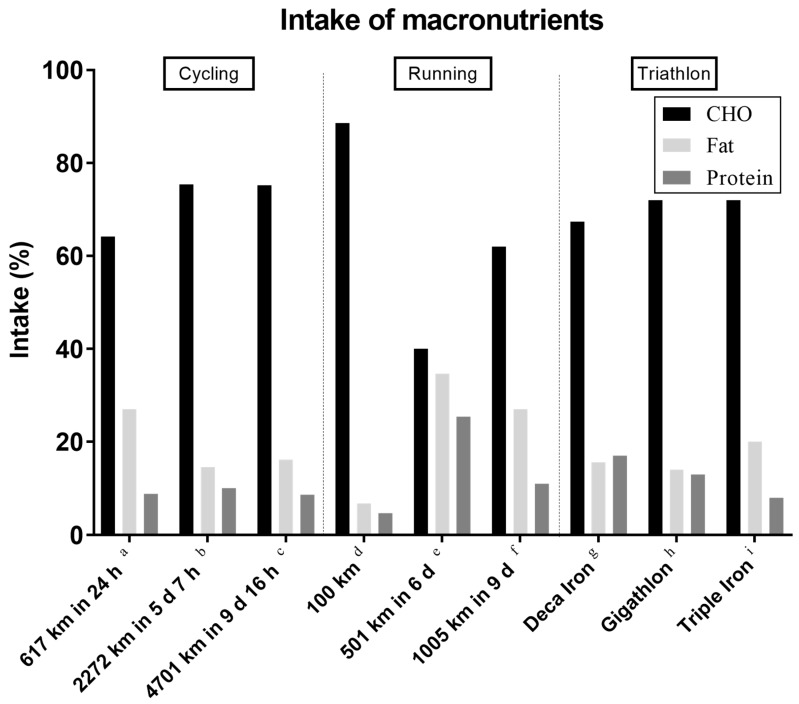
Percentage intake of macronutrients. a = [28], b = [32], c = [11], d = [104], e = [35], f = [105], g = [106], h = [41], i = [40].

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
