# Peer review of "Nutrition in Ultra-Endurance: State of the Art"

_nutrients, 2018, doi:10.3390/nu10121995_

Reviewer 1 Report

Dear authors

I read the paper entitled "Nutrition in ultra-endurance: State of the art" 

Is an interesting topic and the manuscript is well written

Author Response

Reviewer 1

Dear authors

I read the paper entitled "Nutrition in ultra-endurance: State of the art" 

Is an interesting topic and the manuscript is well written

Answer: We thank the expert reviewer for his/her comment.

Reviewer 2 Report

A useful introduction of nutrition during extreme endurance exercise. The use of the illustrations should be reconsidered, some are not even referred to in the text. Only fig 6 appears to contribute to the message of the paper. 

Minor points

the connecting lines in fig 3 do not mean anything

correct typo's in line 211, 236, 248, 252, 285

Author Response

Reviewer 2

A useful introduction of nutrition during extreme endurance exercise. The use of the illustrations should be reconsidered, some are not even referred to in the text. Only fig 6 appears to contribute to the message of the paper. 

Answer: We agree with the expert reviewer and removed all illustrations following the suggestions of reviewer 3, too.

Minor points

The connecting lines in fig 3 do not mean anything

Answer: We agree with the expert reviewer and removed them.

Correct typo's in line 211, 236, 248, 252, 285

Answer: We agree with the expert reviewer and corrected them.

Reviewer 3 Report

My major comment is that the information in Sections 3.1, 4.5 and 4.6 needs to be combined? There seems to be an overlap of information and some inconsistency regarding recommendation of fluid intake.

In the abstract and manuscript is mention of dehydration. It seems that the authors refer to the state of hypohydration as dehyradation is the depletion of fluids from the body. The authors may consider use of the term hypohydration.

L25 and L323. The evidence on “consumption of fat-rich foods” on performance is limited. I would suggest to change “should be” by “may”

L97. Change “~3 more “ to “~3x more “

L110-111. Please clarify “to a fat mass, muscle mass, skeletal muscle mass”. I assume the authors are not considering loss of other muscle mass than skeletal muscle mass”

L111-117. Is swimming also not primarily concentric exercise so it seems than that the loss of skeletal muscle mass is more dependent on the duration of the exercise. Please revise if necessary.

In Fig. 3 I suggest not to connect the datapoints as these are all obsevations from separate studies.

Is figure 4 really needed to make the point of body mass loss? In addition, fig 4 is a 3.3% in body mass loss and not 5% or higher mentioned in L 105.

L154. “shown in several studies [44].” Studies plural but only one reference. Please provide more references.

L161. I suggest to rephrase “pathophysiological aspects” as we are dealing with disease or injury but normal physiological responses to ultra-endurance exercise.

L182. That should be hyponatremia.

L184. Change “typically resulted” to “typically”

Ls 180-183 and L194 seems to be a repeat of the fact that women are at higher risk for hyponatremia. Please revise.

Do we need Figure 6?

L242. Rephrase or clarify “and very tough for the body” .

L243. Change “The Sherman and colleagues protocol was” to “The Sherman and colleagues protocol [75] was”

L246. Change “resent” to “recent”

L249. Change “in this study..” to “In this study [76]…”

Ls 284-285. Effects of protein intake in resistance training studies, refs 83 and 84, may not be comparable with observations in ultra-endurance studies. The authors may want to clarify that.

L288. Change “not” to “no”

Ls 288-291. As there is some evidence on the blunting of training adaptations by intake of vitamin C and E, the authors may want to include that information, e.g. Morrison D, Hughes J, Della Gatta PA, Mason S, Lamon S, Russell AP, Wadley GD. Vitamin C and E supplementation prevents some of the cellular adaptations to endurance-training in humans. Free Radic Biol Med. 2015 Dec;89:852-62. doi: 10.1016/j.freeradbiomed.2015.10.412

Do we need Figure 8?

Do we need Figure 9?

Author Response

Reviewer 3

My major comment is that the information in Sections 3.1, 4.5 and 4.6 needs to be combined?

Answer: We agree with the expert reviewer and combined these sections by moving 4.5 and 4.6 as new subsections 3.2. and 3.3., respectively.

There seems to be an overlap of information and some inconsistency regarding recommendation of fluid intake.

Answer: We agree with the expert reviewer and corrected this issue in the 3.2. part.

In the abstract and manuscript is mention of dehydration. It seems that the authors refer to the state of hypohydration as dehydration is the depletion of fluids from the body. The authors may consider use of the term hypohydration.

Answer: We agree with the expert reviewer and adopted the term hypohydration throughout the abstract and text.

L25 and L323. The evidence on “consumption of fat-rich foods” on performance is limited. I would suggest changing “should be” by “may”

Answer: We agree with the expert reviewer and corrected it as suggested.

L97. Change “~3 more “ to “~3x more “

Answer: We agree with the expert reviewer and corrected it as suggested.

L110-111. Please clarify “to a fat mass, muscle mass, skeletal muscle mass”. I assume the authors are not considering loss of other muscle mass than skeletal muscle mass”

Answer: We agree with the expert reviewer and corrected it.

L111-117. Is swimming also not primarily concentric exercise so it seems than that the loss of skeletal muscle mass is more dependent on the duration of the exercise. Please revise if necessary.

Answer: We agree with the expert reviewer and corrected it.

In Fig. 3 I suggest not to connect the data points as these are all observations from separate studies.

Answer: We agree with the expert reviewer and corrected it.

Is figure 4 really needed to make the point of body mass loss? In addition, fig 4 is a 3.3% in body mass loss and not 5% or higher mentioned in L 105.

Answer: We agree with the expert reviewer and deleted it.

L154. “shown in several studies [44].” Studies plural but only one reference. Please provide more references.

Answer: We agree with the expert reviewer and added literature.

L161. I suggest rephrasing “pathophysiological aspects” as we are dealing with disease or injury but normal physiological responses to ultra-endurance exercise.

Answer: We agree with the expert reviewer and revised this phrase.

L182. That should be hyponatremia.

Answer: We agree with the expert reviewer and corrected it.

L184. Change “typically resulted” to “typically”

Answer: We agree with the expert reviewer and corrected it.

Ls 180-183 and L194 seems to be a repeat of the fact that women are at higher risk for hyponatremia. Please revise.

Answer: We agree with the expert reviewer and deleted the second one.

Do we need Figure 6?

Answer: We agree with the expert reviewer and deleted it.

L242. Rephrase or clarify “and very tough for the body” .

Answer: We agree with the expert reviewer and changed to „stressing“.

L243. Change “The Sherman and colleagues protocol was” to “The Sherman and colleagues protocol [75] was”

Answer: We agree with the expert reviewer and corrected it.

L246. Change “resent” to “recent”

Answer: We agree with the expert reviewer and corrected it.

L249. Change “in this study..” to “In this study [76]…”

Answer: We agree with the expert reviewer and corrected it.

Ls 284-285. Effects of protein intake in resistance training studies, refs 83 and 84, may not be comparable with observations in ultra-endurance studies. The authors may want to clarify that.

Answer: We agree with the expert reviewer and deleted it.

L288. Change “not” to “no”

Answer: We agree with the expert reviewer and corrected it.

Ls 288-291. As there is some evidence on the blunting of training adaptations by intake of vitamin C and E, the authors may want to include that information, e.g. Morrison D, Hughes J, Della Gatta PA, Mason S, Lamon S, Russell AP, Wadley GD. Vitamin C and E supplementation prevents some of the cellular adaptations to endurance-training in humans. Free Radic Biol Med. 2015 Dec;89:852-62. doi: 10.1016/j.freeradbiomed.2015.10.412

Answer: We agree with the expert reviewer and added this aspect.

Do we need Figure 8?

Answer: We agree with the expert reviewer and removed it as well as its reference within the text.

Do we need Figure 9?

Answer: We agree with the expert reviewer and removed it as well as its reference within the text.

Round  2

Reviewer 1 Report

The authors has been many modification as requested by editor. 

I think the paper has been improved, but I'm not able to establish the level of english language.

Author Response

Comments and Suggestions for Authors

The authors has been many modification as requested by editor. 

I think the paper has been improved, but I'm not able to establish the level of english language.

Answer: We thank the expert reviewer for his/her comments, no further changes are required.

Reviewer 2 Report

In my opinion the manuscript is improved considerably.
It is not very imaginitive, but rather a useful collection of data for outsiders.

Author Response

Comments and Suggestions for Authors

In my opinion the manuscript is improved considerably.
It is not very imaginitive, but rather a useful collection of data for outsiders.

Answer: We thank the expert reviewer for his/her comments, no further changes are required.

Reviewer 3 Report

I have no further comments. Thanks for replying to all my previous comments.

Author Response

Comments and Suggestions for Authors  

I have no further comments. Thanks for replying to all my previous comments.

Answer: We thank the expert reviewer for his/her comments, no further changes are required.

Round  3

Reviewer 3 Report

I agree with the editor's comments and the paper still needs major revision.

Ls 15-16. The abstract needs to be clear what the exercise duration is for which there is evidence that there is a decrease in muscle mass as the abstract now suggests that just 6 hr of exercise leads to a decrease in muscle mass. This evidence needs to be presented in the paper clearly to justify this statement in the abstract.

L45. Clarify that exercise-induced hyponatremia is a risk.

L72. Please revise “To counterbalance this energy consumption, an optimal nutrition should ensure adequate energy intake [14]. “ I guess counterbalancing increased energy expenditure.

L75. Refs 15 and 16 do not provide information on decreases in muscle mass. Please revise. In addition, refs 30, 31, and 33 reveal inconsistent findings regarding changes in muscle mass. This needs to be recognized. Also, the manuscript should be clear in the text on what the exercise duration needs to be in order to expect decreases in muscle mass.

L86. “swimming induces a larger energy deficit”. Please clarify whether this is due to hampered energy intake.

Ls 118-120. Statement indicates no changes in muscle mass in contrast with other statements in the manuscript.

Ls 126-128. “Therefore, the decrease of body mass, fat and skeletal muscle mass during training of ultra-endurance athletes would be expected to be less than in ultra-endurance races.” This statement seems to suggest that 8 hr running weekly may result in skeletal muscle mass. Evidence for this statement needs to be provided. It remains very speculative.

L177. Change “max fluid” to “maximum fluid”

It seems sections 3.1 and 3.2 and maybe 3.3. can be combined. In addition, there seems to be some repeat of information in those sections.

L268. Train low run run high is not mentioned in Ref 84. In addition, it refers to practice to enhance training adaptations. Please revise. In addition, the paper would benefit from more indepth coverage of such recent nutritional practices to enhance endurance performance and training adaptations, also known as carbohydrate periodization. Is training with low glycogen common practice for ultra-endurance athletes?

L269. Sentence starting with “Nevertheless” is not clear. Please revise.

Ls 272-273. Please clarify/revise “ultra-endurance sports presented increased demands in fat intake considering the length of the races.”

L281. “The role of fat consumption has been studied during training…”.Please justify with studies and outcomes.

Author Response

Please, find our answers to the specific questions below and the changes in the text highlighted in blue.

I agree with the editor's comments and the paper still needs major revision.

Ls 15-16. The abstract needs to be clear what the exercise duration is for which there is evidence that there is a decrease in muscle mass as the abstract now suggests that just 6 hr of exercise leads to a decrease in muscle mass. This evidence needs to be presented in the paper clearly to justify this statement in the abstract.

Answer: We agree with the expert reviewer and revised this part in the abstract (in events such as 24h swimming, 6-day cycling or 17-day running”) and in the section 2.2.we added “For instance, the largest decrease of skeletal muscle mass was observed in events such as 24h swimming, 6-day cycling or 17-day running (Figure 3)”.

L45. Clarify that exercise-induced hyponatremia is a risk.

Answer: We agree with the expert reviewer and clarified it adding “For instance, an increased fluid intake compared to fluid loss might lead to exercise-induced hyponatremia - which is associated with hyponatraemic encephalopathy, altered mental status, collapse, seizure, coma and death - during prolonged races [3].”.

L72. Please revise “To counterbalance this energy consumption, an optimal nutrition should ensure adequate energy intake [14]. “ I guess counterbalancing increased energy expenditure.

Answer: We agree with the expert reviewer and revised it to “To balance the increased energy expenditure, ...”.

L75. Refs 15 and 16 do not provide information on decreases in muscle mass. Please revise. In addition, refs 30, 31, and 33 reveal inconsistent findings regarding changes in muscle mass. This needs to be recognized. Also, the manuscript should be clear in the text on what the exercise duration needs to be in order to expect decreases in muscle mass.

Answer: We agree with the expert reviewer and revised it. We recognized the discrepancy among studies (“Although most of the research conducted in ultra-endurance races reported a decrease of body mass across the races, inconsistencies in the findings regarding changes in fat and skeletal muscle mass among studies [30,31,33] should be attributed to race specific characteristics (e.g. duration and mode of exercise) and inter-individual variability (i.e. responders and non-responders) [15].”). For muscle mass, please read our first answer.

L86. “swimming induces a larger energy deficit”. Please clarify whether this is due to hampered energy intake.

Answer: We agree with the expert reviewer and clarified it.

Ls 118-120. Statement indicates no changes in muscle mass in contrast with other statements in the manuscript.

Answer: We agree with the expert reviewer and deleted it.

Ls 126-128. “Therefore, the decrease of body mass, fat and skeletal muscle mass during training of ultra-endurance athletes would be expected to be less than in ultra-endurance races.” This statement seems to suggest that 8 hr running weekly may result in skeletal muscle mass. Evidence for this statement needs to be provided. It remains very speculative.

Answer: We agree with the expert reviewer and revised it („So far, in contrast to the abovementioned observations of larger energy expenditure than intake during ultra-endurance races [25-28], a negative energy balance has not been reported during training of ultra-endurance athletes, and thus, it might be assumed that changes of body mass, fat and skeletal muscle mass during training might be trivial.“).

L177. Change “max fluid” to “maximum fluid”

Answer: We agree with the expert reviewer and corrected it.

It seems sections 3.1 and 3.2 and maybe 3.3 can be combined. In addition, there seems to be some repeat of information in those sections.

Answer: We agree with the expert reviewer and combined all sub-sections and double-checked its content.

L268. Train low run high is not mentioned in Ref 84. In addition, it refers to practice to enhance training adaptations. Please revise. In addition, the paper would benefit from more in-depth coverage of such recent nutritional practices to enhance endurance performance and training adaptations, also known as carbohydrate periodization. Is training with low glycogen common practice for ultra-endurance athletes?

Answer: We agree with the expert reviewer and revised this part. First, we changed the reference for [84] (the new one is Jeukendrup, A.E. Periodized nutrition for athletes. Sports Medicine (Auckland, N.Z.) 2017, 47, S51-63.). Second, we developed further the concept of carbohydrate periodization; however, limited information existed on nutritional practice during training in ultra-endurance („The recently developed concept of periodized nutrition has been defined as ‘planned, purposeful, and strategic use of specific nutritional interventions to enhance the adaptations targeted by individual exercise sessions or periodic training plans, or to obtain other effects that will enhance performance longer term’ [84]. Unfortunately, the CHO intake in ultra-endurance athletes has been studied mostly during race, whereas scarce information existed with regards to nutritional behavior during training.“).

L269. Sentence starting with “Nevertheless” is not clear. Please revise.

Answer: We agree with the expert reviewer and revised it adding “...since the periodic consumption of low CHO did not reveal superior effects on performance of endurance athletes compared to high CHO intake”.

Ls 272-273. Please clarify/revise “ultra-endurance sports presented increased demands in fat intake considering the length of the races.”

Answer: We agree with the expert reviewer and revised it (“...and that larger amount of fat than CHO was stored in human body[86].”).

L281. “The role of fat consumption has been studied during training…”.Please justify with studies and outcomes.

Answer: We agree with the expert reviewer and explained it revising this sentence (“The role of fat consumption has been studied during training [82,92] and race [77,93], where a nutritional strategy in its intake was to increase fat acid oxidation for a given exercise intensity and meet the increased caloric demands, respectively.”).
